# Yeast Sphingolipid-Enriched Domains and Membrane Compartments in the Absence of Mannosyldiinositolphosphorylceramide

**DOI:** 10.3390/biom10060871

**Published:** 2020-06-06

**Authors:** Andreia Bento-Oliveira, Filipa C. Santos, Joaquim Trigo Marquês, Pedro M. R. Paulo, Thomas Korte, Andreas Herrmann, H. Susana Marinho, Rodrigo F. M. de Almeida

**Affiliations:** 1Centro de Química Estrutural, Faculdade de Ciências, Universidade de Lisboa, 1749-016 Lisbon, Portugal; abdoliveira@fc.ul.pt (A.B.-O.); fpsantos@fc.ul.pt (F.C.S.); jmtmarques@fc.ul.pt (J.T.M.); hsmarinho@fc.ul.pt (H.S.M.); 2Centro de Química Estrutural, Instituto Superior Técnico, 1049-001 Lisbon, Portugal; pedro.m.paulo@tecnico.ulisboa.pt; 3Department of Biology, Molecular Biophysics, IRI Life Sciences, Humboldt-Universität zu Berlin, 10115 Berlin, Germany; thomas.korte@rz.hu-berlin.de (T.K.); andreas.herrmann@rz.hu-berlin.de (A.H.)

**Keywords:** *Saccharomyces cerevisiae*, membrane compartments, sphingolipids, Pma1p, Can1p, fluorescence lifetime imaging microscopy (FLIM), fungal plasma membrane, inositolphosphorylceramides, fluorescence spectroscopy, giant unilamellar vesicles (GUVs)

## Abstract

The relevance of mannosyldiinositolphosphorylceramide [M(IP)_2_C] synthesis, the terminal complex sphingolipid class in the yeast *Saccharomyces cerevisiae*, for the lateral organization of the plasma membrane, and in particular for sphingolipid-enriched gel domains, was investigated by fluorescence spectroscopy and microscopy. We also addressed how changing the complex sphingolipid profile in the plasma membrane could influence the membrane compartments (MC) containing either the arginine/ H^+^ symporter Can1p (MCC) or the proton ATPase Pma1p (MCP). To achieve these goals, wild-type (*wt*) and *ipt1*Δ cells, which are unable to synthesize M(IP)_2_C accumulating mannosylinositolphosphorylceramide (MIPC), were compared. Living cells, isolated plasma membrane and giant unilamellar vesicles reconstituted from plasma membrane lipids were labelled with various fluorescent membrane probes that report the presence and organization of distinct lipid domains, global order, and dielectric properties. Can1p and Pma1p were tagged with GFP and mRFP, respectively, in both yeast strains, to evaluate their lateral organization using confocal fluorescence intensity and fluorescence lifetime imaging. The results show that *IPT1* deletion strongly affects the rigidity of gel domains but not their relative abundance, whereas no significant alterations could be perceived in ergosterol-enriched domains. Moreover, in these cells lacking M(IP)_2_C, a clear alteration in Pma1p membrane distribution, but no significant changes in Can1p distribution, were observed. Thus, this work reinforces the notion that sphingolipid-enriched domains distinct from ergosterol-enriched regions are present in the *S. cerevisiae* plasma membrane and suggests that M(IP)_2_C is important for a proper hydrophobic chain packing of sphingolipids in the gel domains of *wt* cells. Furthermore, our results strongly support the involvement of sphingolipid domains in the formation and stability of the MCP, possibly being enriched in this compartment.

## 1. Introduction

The plasma membrane of fungi contains a multitude of actively maintained membrane compartments, which have distinct protein markers, are stable in space and time, and are responsible for specific physiological functions [1,2,3]. Despite the growing interest in these compartments, the mechanism of their formation, their lipid composition, and their interrelations are still largely unknown [4].

In yeast and other fungi, as well as in plants, many plasma membrane proteins display slow lateral diffusion or are immobile, when compared to vertebrate biological membranes or artificial fluid lipid bilayers [2]. In some cases, this low mobility is dependent on plasma membrane sterols and sphingolipids [1,3,5,6,7,8]. In this regard, sphingolipid-enriched microdomains seem to be of particular relevance [9,10,11]. These domains found in the plasma membrane of *Saccharomyces cerevisiae* differ from the prototypical mammalian lipid raft both in composition, given they are ergosterol-depleted, and biophysically, as they are highly rigid gel domains [11]. Hence, the study of these domains is particularly important, as they may explain fundamental differences between fungal and mammalian membranes [9], providing potential targets that can be explored concerning antifungal drug development [12].

In the plasma membrane of yeast *S. cerevisiae* two major non-overlapping compartments can be found: The membrane compartment containing Can1p (MCC), an arginine permease, and the membrane compartment containing Pma1p (MCP), the plasma membrane H^+^-ATPase. MCC is currently designated MCC/eisosome, as its characteristic furrow-like invaginating shape is stabilized by a large cytosolic protein scaffold, the eisosome, at the bottom of the MCC furrow [2,13,14]. A typical *S. cerevisiae* cell in exponential phase contains around 1.5–2.5 such invaginations per μm^2^ of cell surface [8,13] covering roughly 25% of the membrane area. Fluorescence labeling with the 3′-β-sterol-binding filipin suggested that the MCC/eisosome is ergosterol-enriched [15]. The deletion of proteins that are part of the MCC/eisosome, or are associated with it, compromises the formation of intact MCC/eisosome domains and influences ergosterol distribution in the plasma membrane [2]. Some of the well-known functions of the MCC/eisosome include the sensing of local membrane disturbances and deformations, e.g., due to mechanical or osmotic stress [16], involvement in membrane adaptation to other types of stress [17,18,19,20], regulation of lipid homeostasis, in particular sensing sphingolipid levels [2], and protection against endocytosis/starvation [3]. Notably, membrane potential has been suggested to modulate the organization of MCC domains. Readily after membrane depolarization, proteins including Can1p are no longer confined to the MCC, acquiring a more homogeneous distribution in the plasma membrane [15,21]. Concomitantly, sphingolipid-enriched microdomains dissipate [22]. Thus, despite the stability of MCC and the low mobility of Can1p inside the MCC, its distribution in the plasma membrane is dynamic and regulated.

The MCP was first identified in *S. cerevisiae* by confocal fluorescence microscopy as a network-like compartment containing Pma1p [23], the most abundant plasma membrane protein in fungi [24]. This protein is responsible for the creation and maintenance of membrane potential of fungal cells, as well as regulation of the intracellular pH and ion concentration [25]. More recently, it has been suggested to be of relevance for signaling [3]. The electrochemical gradient established by Pma1p drives nutrient uptake, namely by symporters localized at the MCC, such as Can1p, establishing an important functional relation between MCP and MCC.

On the other hand, in *S. cerevisiae* [26,27] as well as in *Cryptococcus neoformans* [28,29], alterations of sphingolipid metabolic balance affect the oligomerization, traffic, activity, and plasma membrane stability of Pma1p [30,31,32,33]. These results point to an intimate relation between the MCP and the sphingolipids. Pma1p has relatively long hydrophobic transmembrane segments and is one of the yeast plasma membrane constituents with the longest residence time [34]. Thus, this compartment seems to be a good candidate to host sphingolipids. With very long hydrocarbon chains, they exhibit tendency to form highly ordered gel domains, and relatively low levels of ergosterol [15] are putatively found in the MCP. However, whether sphingolipids are major lipid components of those domains is also unproved, as is the sphingolipid enrichment in the MCC. Moreover, the labelling of sphingolipids is not straightforward, and only indirect evidence is available to support that hypothesis.

Based on the structure of their headgroup, complex sphingolipids are classified in three types in yeast, inositolphosphorylceramide (IPC), MIPC (mannosylinositolphosphorylceramide), and M(IP)_2_C (mannosyldiinositolphosphorylceramide). M(IP)_2_C is the most abundant complex sphingolipid in *S. cerevisiae*, regardless of whether the whole cell or specifically the plasma membrane is considered [35,36]. Still, MIPC is also present in the plasma membrane of wild-type (*wt*) *S. cerevisiae* cells. The formation of terminal complex sphingolipid M(IP)_2_C from MIPC is catalyzed by an inositol phosphotransferase encoded by the *IPT1* gene [35,36]. In *ipt1*Δ cells, despite the major changes in sphingolipid composition, that are characterized by the lack of M(IP)_2_C and its replacement by MIPC, the sterol composition is similar to that of *wt* cells [35]. Some studies suggest that this gene deletion may enhance resistance to both nystatin and miconazole [35], and confers markedly higher resistance to zymocin [37], syringomycin E [38], and the plant defensin DmAMP1 produced by *Dahlia merckii* seeds [39]. The authors of the latter study proposed that the antifungal peptide targets membrane patches containing M(IP)_2_C, although the specific contribution of each complex sphingolipid for the organization of the plasma membrane is still unclear.

In this work, in an attempt to clarify some of the issues described above, we have carried out a thorough biophysical characterization of the plasma membrane in *S. cerevisiae ipt1*Δ versus *wt* cells, addressing both lipid domains and membrane compartments. We found that the plasma membrane of *ipt1*Δ cells has more compact sphingolipid-enriched domains, whereas dielectric properties that are more specifically modulated by ergosterol remained essentially unchanged. Moreover, we present evidence that in *ipt1*Δ cells, where M(IP)_2_C is absent, the plasma membrane distribution of Pma1p is much more affected than that of Can1p.

## 2. Materials and Methods

### 2.1. Materials and Strains

Table 1 lists the *S. cerevisiae* strains used in this work. The *wt* and *ipt1*Δ strains were transformed using the lithium acetate procedure [40], with the integrative plasmid Ylp211-CAN1-GFP (*URA3*) or Ylp128-PMA1-mRFP (*LEU2*) kindly provided by Dr. Guido Grossmann and Professor Widmar Tanner (University of Regensburg, Germany).

Yeast extract, bactopeptone, yeast nitrogen base, and bacto-agar were purchased from Difco (Hampton, NH, USA). Amino acids, nucleotides and glucose used in the synthetic complete (SC) medium, sucrose, and organic solvents (of analytical grade for lipid extraction and spectroscopic grade for lipids and probes stock solutions) were obtained from Merck (Darmstadt, Germany). *Trans*-parinaric acid (*t*-PnA) was purchased from Santa Cruz Biotech. (Santa Cruz, CA, USA), 1,6-diphenylhexatriene (DPH) from Invitrogen (Carlsbad, CA, USA), 4-[2-[6-(dioctylamino)-2-naphthalenyl]ethenyl]-1-(3-sulfopropyl)-pyridinium (di-8-ANEPPS) was obtained from Biotium (Fremont, CA, USA), 1,2-dioleoyl-*sn*-glycero-3-phosphoethanolamine-*N*-(lissamine rhodamine B sulfonyl) (Rhod-DOPE) from Avanti Polar Lipids (Alabaster, AL, USA), and Ludox (TM-50 colloidal silica, 50 wt.% suspension in water) acquired from Sigma-Aldrich (St. Louis, MO, USA).

The fluorescent probes were quantified spectrophotometrically using the respective molar absorption coefficients: *t*-PnA,$\;\varepsilon_{ethanol}^{299.4\;nm}$ = 89,000 M^−1^cm^−1^; DPH, $\varepsilon_{methanol}^{350\;nm}$ = 88,000 M^−1^cm^−1^; di-8-ANEPPS, $\varepsilon_{methanol}^{498\;nm}$ = 37,000 M^−1^cm^−1^, Rhod-DOPE, $\varepsilon_{chloroform}^{559\;nm}$ = 95,000 M^−1^cm^−1^ [41].

### 2.2. Media and Growth Conditions

*S. cerevisiae* cells were inoculated at an A_600nm_ of 0.15 and cultured in SC medium (2% (*w*/*v*) glucose, 0.68% (*w*/*v*) yeast nitrogen base and amino acid composition given in [42]), except when specifically stated and for microscopy experiments using di-8-ANEPPS and Rhod-DOPE as membrane probes. In these cases, a similar growth procedure was performed using a culture medium containing 1% (*w*/*v*) yeast extract, 2% (*w*/*v*) peptone and 2% (*w*/*v*) glucose (YPD). Yeast cells were grown at 30 °C and 160 rpm overnight and reinoculated into the culture medium at an A_600nm_ of ~0.15. Upon reaching an A_600nm_ of 0.6, after a 5–6 h incubation, i.e., in the mid-exponential phase, cells were harvested.

### 2.3. Plasma Membrane Isolation

The plasma membrane isolation was performed as described in [43] with minor adjustments. Briefly, cells were collected by centrifugation (5 min, 5000× *g*) and washed twice with 0.4 M sucrose in buffer A (25 mM imidazole HCl, pH 7.0) at 3500× *g* for 10 min. Harvested cells were resuspended in buffer A containing sucrose at 0.4 M and a mixture of protease inhibitors (Halt™ protease inhibitor cocktail (100×) from Thermo Scientific (Waltham, MA, USA)), and lysed gently with glass beads. To obtain a crude membrane extract, lysed cells were centrifuged at low speed (530× *g*) for 20 min, the supernatant was recovered and centrifuged at 22,000× *g* for 30 min. For the total lipid extracts preparation, the pellet containing the crude membrane extract was stored, while for the plasma membrane isolation it was resuspended in 2 mL of buffer A containing the protease inhibitor mixture. Then, it was applied on the top of a 10 mL discontinuous sucrose gradient (consisting of three layers of 2.25, 1.65, and 1.1 M sucrose in buffer A). The gradient centrifugation was carried out in a CP80NX Hitachi ultracentrifuge (Berkshire, United Kingdom) for 18 h at 80,000× *g*. Pure plasma membrane fractions were obtained from the interface between the 2.25 and 1.65 M sucrose layers and then they were resuspended in either buffer A (for lipid extraction) or B (for fluorescence analysis; 100 mM NaH_2_PO_4_.H_2_O, 100 mM NaCl, 1 mM EDTA, pH 7.4) and pelleted by centrifugation at 30,000× *g* for 40 min.

### 2.4. Lipid Extraction

The extraction of the total lipids of the cell and of the plasma membrane was performed using a slightly modified Bligh–Dyer method [44]. The isolated plasma membrane (IPM) and the crude membrane extracts were separately incubated with 1 mL of buffer A and 19 mL of chloroform:methanol (1:2, *v*/*v*) at 20 °C with shaking at 140 rpm for 1 h. Chloroform was further added in order to have a chloroform:methanol ratio of 1:1 (*v*/*v*). The same volume of a 0.04% (*w*/*v*) CaCl_2_ solution was added to the mixture, which was then vigorously stirred for 1 min (while relieving the pressure). Once two distinct phases were visible, both the organic phase and the interface were collected separately. The whole process was then repeated for the interface, from which the organic phase was collected, added to the previous one, dried under a stream of nitrogen and then stored at −20 °C. In plasma membrane suspensions and in lipid extracts, total phospholipid concentration was determined by phosphorus analysis [45]. Ergosterol molar proportion was assumed to be ca. 2:1 (phospholipid:ergosterol) [19,46]. Before use, total lipid extracts were redissolved in chloroform/methanol (2:1, *v*/*v*) and the adequate volume was taken to obtain suspensions of multilamellar vesicles (MLVs) in buffer B, at a final lipid concentration of ca. 0.3 mM (see [11] for further details).

### 2.5. Giant Unilamellar Vesicles (GUVs) Preparation

GUVs were prepared by the electroformation method of Angelova et al. [47]. For GUV preparation, a 1 mM solution of the lipids extracted from yeast plasma membrane in chloroform: methanol (2:1, *v*/*v*) was used. The adequate volume of probe (di-8-ANEPPS and/ or Rhod-DOPE) stock solution was mixed to obtain a final lipid/probe ratio of 1:500 (mol:mol) and a chloroform:methanol ratio of (2:1, *v*/*v*). Then, 25 µL of this mixture, previously homogenized, was spread with a gas-tight syringe on the titanium plates of the GUV chamber. To remove any traces of organic solvent, the plates were placed on a vacuum desiccator for 30 min. The titanium chambers were assembled and filled with 700 µL of a 200 mM sucrose solution with sodium azide 15 mM, previously heated to 60 °C in a heat block, sealed, connected to the signal generator and placed in the heat block maintaining the system at 60 °C. A 10 Hz sinusoidal wave with a peak-to-peak amplitude of 0.4 V was initially applied and the voltage was increased to 2.0 V and kept overnight. The sinusoidal signal was then changed to 4 Hz frequency and 2.6 V voltage for 20 min. The vesicle suspension was kept at room temperature (24 °C) in the dark until use. For GUV visualization, 100 µL of each GUV suspension and 150 µL of glucose solution 200 mM were added to a well of an eight-well plastic plate from Ibidi^®^ with glass like coverslip bottom.

### 2.6. Fluorescence Spectroscopy Measurements and Data Analysis

Fluorescence measurements were performed with a Fluorolog Model 3.22 spectrofluorimeter from Horiba Jobin Yvon (Villeneuve D’ascq, France) at 24.0 ± 1.0 °C in a sample compartment with temperature-control and magnetic stirring.

Harvested cells (*wt* and *ipt1*Δ) were washed twice with sterile water and then suspended in buffer B. Before starting the measurements, the probes DPH or *t*-PnA were added to the cell suspension to a final concentration of 2 µM and incubated at 24 °C (room temperature) for 20 and 5 min, respectively [11].

For fluorescence measurements, the IPM suspensions were incubated with *t*-PnA, DPH or di-8-ANEPPS for more than 5 min, 20 min, and 1 h respectively, at room temperature, to a final ratio of probe/lipid of 1/100, for DPH and *t*-PnA, and 1/200 for di-8-ANEPPS. The MLV suspensions made from total lipid extracts were labelled with DPH at a probe:phospholipid ratio of 1/200. following a procedure previously described [48].

For fluorescence in the steady-state, the excitation and emission wavelengths were set to, respectively, 320 and 404 nm for *t*-PnA and 358 and 430 nm for DPH. The value of steady-state fluorescence anisotropy, 〈r〉, was determined through Equation (1):(1)$$\langle r\rangle =\frac{\left(I_{\mathrm{VV}\;}-\;G\;\times {\;I}_{\mathrm{VH}}\right)}{\left(I_{\mathrm{VV}}+2G\;\times {\;I}_{\mathrm{VH}}\right)}$$
in which the subscripts indicate the orientation of the excitation and emission polarizers, where V and H correspond to vertical and horizontal orientations and *G* is a correction for the instrument different sensitivity to each light polarization plane. An adequate blank was subtracted from each intensity reading.

Time-resolved fluorescence measurements were carried out by the single photon timing technique, using a nanoLED N-320 from Horiba Jobin Yvon (Villeneuve D’ascq, France) for the excitation of *t*-PnA, and 404 nm as the emission wavelength. The scattering agent used to obtain the instrument response function (IRF) was Ludox^®^. The experimental fluorescence intensity decays were analyzed with the software TRFA^®^ version 1.4 (Minsk, Belarus). To completely separate the emission of the probe from the autofluorescence of the cells, a method of global analysis was used [49]. The fluorescence intensity decays of the probe were described by a multiexponential model according to Equation (2):(2)$$I\left(t\right)=\sum_{i=1}^{n}\alpha_{i}\exp(-\frac{t}{\tau_{i}}).$$

After subtracting the contribution of the cells autofluorescence to the pre-exponential factor in each component *i* with lifetime $\tau_{i}$, and after the thus obtained pre-exponential factors were normalized to obtain their relative amplitude $\alpha_{i}$. Equation (3) was used to calculate the mean fluorescence lifetime (intensity weighted):(3)$$\langle \tau \rangle =\frac{\sum \alpha_{i}\tau_{i}^{2}}{\sum \alpha_{i}\tau_{i}},$$
and Equation (4) was used to compute the amplitude-weighted mean fluorescence lifetime:(4)$$\bar{\tau }=\sum_{i=1}^{n}\alpha_{i}\tau_{i}.$$

The criteria for good quality of the fitting were the reduced global χ^2^ close to 1 and the random distributions of weighted residuals and of residuals autocorrelation (see Appendix A).

Changes in membrane dipole potential were evaluated in IPM suspensions using the ratio of the steady-state fluorescence intensity values obtained from excitation of di-8-ANEPPS at 420 nm and at 520 nm, both with emission at 635 nm [50,51].

### 2.7. Fluorescence Intensity and Lifetime Imaging by Confocal Microscopy

#### 2.7.1. GUV and Living Cells Labeled with Di-8-ANEPPS or Rhod-DOPE

For fluorescence lifetime imaging microscopy (FLIM) with di-8-ANEPPS, after cell harvesting, the cellular suspension was labeled at room temperature (24 °C) with the probe at 1 µM, which was added from a concentrated methanol stock solution. After a 5 min incubation with the probe, the cells were centrifuged and suspended in buffer B.

Intensity images as well as FLIM measurements were taken in the confocal mode using an Olympus Fluoview 1000 (Olympus, Tokyo, Japan) upgraded with a LSM time-resolved kit (PicoQuant GmbH, Berlin, Germany) and an oil immersion objective, ×60 (1.35 N.A.). For intensity images, Di-8-ANEPPS was excited by a 488 nm line of a multiline argon laser and its emission detected in the range of 565–665 nm. Rhod-DOPE intensity images were acquired with excitation at 559 nm using a line of a multiline argon laser and detection in the range of 590–690 nm. Images with a frame size of 512 × 512 pixels were acquired. For FLIM measurements, di-8-ANEPPS was excited at 483 nm and Rhod-DOPE at 560 nm using pulsed-laser diodes and detected by a single photon avalanche photodiode (SPAD) using a 620 ± 30 nm bandpass filter. FLIM images were acquired for 60 s (45 frames with an average photon count rate of 2–4 × 10^4^ count/s). Analysis of FLIM data was performed using the software from PicoQuant, SymPhoTime 64, (version 2.5), considering the convolution of the fluorescence decays with the IRF. The time distribution of photon counts in each pixel retrieved from the regions of interest were combined into a single decay curve which was analyzed by using a 2 exponentials model (Equation (2) with *i* = 2), judging the fitting quality from the distribution of the residuals and reduced χ^2^ value. The intensity-weighted mean fluorescence lifetime τ and the amplitude weighted mean fluorescence lifetime $\bar{\tau }$ were calculated using Equations (3) and (4). The results are averaged over more than 50 GUV from at least 3 independent experiments and more than 400 cells from 4 independent experiments. The fluorescence lifetime of the probe was independent of the incubation time.

#### 2.7.2. Yeast Living Cells Tagged with GFP and mRFP

FLIM measurements as well as confocal images of harvested *S. cerevisiae wt* and *ipt1*Δ cells with GFP-tagged Can1p and mRFP-tagged Pma1p were performed on a time-resolved confocal fluorescence microscope, model MicroTime 200, from PicoQuant GmbH (Berlin, Germany), with details of the equipment setup described in [52].

The point-spread function of the setup was measured using fluorescent beads of size 100 nm, which afforded a lateral resolution of about 400 nm. Intensity images were obtained concomitantly with FLIM measurements. GFP and mRFP were excited at 480 nm using a pulsed-laser diode and detected by a SPAD using a the longpass filter above 510 nm. The images were obtained with a scanning resolution of 0.156 µm/pixel. All images were acquired with an excitation power of ca. 0.54 kW/cm^2^ and further processed by creating a mask in which the cell membrane regions were visually selected using the region-of-interest (ROI) tools of SymPhoTime program, version 5.3.2.2. The fluorescence decays corresponding to the regions identified as cell membranes were obtained from the histograms of photon arrival times concerning only the pixels contained in these regions. Here, the same criteria as in Section 2.7.1. for the analysis of the FLIM data of proteins were employed. The results are averaged over ca. 400 cells from 3 independent experiments.

Confocal images were used to assess the distribution of Can1p-GFP and Pma1p-mRFP in the plasma membrane of yeast living cells. For numerical data extraction a line profile of the fluorescence intensity of the plasma membrane was obtained using Image Pro-Plus v6.0 and the fluorescence heterogeneity, i.e., the fraction of pixels which have intensity higher or lower than the average plus or minus 10%, respectively (or plus or minus the standard deviation—both calculations yield the same relative results when comparing proteins or strains). It reflects to which extent the fluorescence intensity is concentrated in specific pixels and is independent on the total fluorescence intensity (see Appendix A). The heterogeneity was determined for the plasma membrane of each cell. For each transformed yeast strain, 4 independent experiments were performed, and more than 190 cells were analyzed.

### 2.8. Statistical Analysis

The results are presented as mean ± standard deviation (S.D.), unless stated otherwise, and the sample dimension and number of biologically independent replicates are given with the results. The statistical significance was determined using Student’s *t*-test. Mean values were considered significantly different for *p* values below or equal to 0.05.

## 3. Results

### 3.1. Sphingolipid-Enriched Domains are more Compact in *ipt1*Δ Cells

To understand how sphingolipid-enriched gel domains may change with an alteration of the sphingolipid profile, the plasma membrane of *S. cerevisiae* living cells was labelled with *t*-PnA. The fluorescence properties of this probe, especially the time-resolved, are sensitive to changes in the amount and composition of highly ordered gel domains, into which it partitions very favorably in membrane model systems, undergoing a concomitant increase of its fluorescence quantum yield [48,53,54]. Additionally, the long lifetime component in *t*-PnA fluorescence intensity decay can be used as a fingerprint for gel domains when longer than 30 ns [11].

The fluorescence intensity decay of *t*-PnA incorporated in the plasma membrane of *S. cerevisiae* cells is properly described by four exponentials, where the long component lifetime (τ_long_) gives information about the relative degree of hydrophobic packing of gel domains and its amplitude (normalized pre-exponential) about their relative abundance [55]. For *wt* cells, the long lifetime component was 40.9 ± 0.4 ns, a value similar to that obtained in our previous work (~41 ns) [11], while for *ipt1*Δ cells it was 45.1 ± 2.4 ns (Figure 1–τ_long_). Since this parameter is higher than 30 ns for both strains, it reveals the presence of gel domains in the plasma membrane [11]. Moreover, both the long-lifetime component and the intensity weighted mean fluorescence lifetime (Figure 1—〈τ〉) were even longer in *ipt1*Δ cells than *wt* cells. Similar results were obtained for cells grown in YPD medium (Appendix A).

Regarding the relative abundance of the sphingolipid-enriched domains, the amplitude of the long lifetime component was somewhat lower in *ipt1*Δ cells, but the differences were not statistically significant. Taken together the results of packing and abundance of the sphingolipid-enriched domains contribute to the identical values obtained for the steady state fluorescence anisotropy of *t*-PnA in living cells or IPM of *S. cerevisiae wt* and *ipt1*Δ cells (Appendix A).

### 3.2. Membrane Fluidity Is Altered in *ipt1*Δ Cells

To further understand how the plasma membrane responds to alterations in the sphingolipid profile, the membrane of *S. cerevisiae* was labelled with DPH. Since DPH partitions equally throughout different lipid phases, it is sensitive to the membrane global properties, rather than to a particular type of domain [56,57], with the notable exception of poorly hydrated ceramide-rich phases [54]. DPH steady-state fluorescence anisotropy, 〈*r*〉, is a well-established parameter to report alterations in the global fluidity of the plasma membrane [11,58,59]. The fluorescence anisotropy of DPH changes linearly with composition (and temperature) along the gel/ liquid disordered (*l_d_*) and *l_d_*/ liquid ordered (*l_o_*) phase coexistence region, i.e., responds linearly to the fraction of each phase, in several binary and ternary lipid systems [12,54,58], including in the 1-palmitoyl-2-oleoyl-*sn*-glycero-3-phosphocholine (POPC)/ergosterol binary system [60]. Moreover, in the POPC/phytoceramide (the yeast sphingolipid backbone) system, the fluorescence anisotropy and intensity of DPH behave similarly to typical gel/ *l*_d_ systems in the region of coexistence of *l_d_* and a phytoceramide-rich gel phase where the hydrophobic packing is comparable to the one found for the gel domains in yeast [48]. All these studies support the use of DPH anisotropy as a reporter of global membrane fluidity in yeast membranes.

As can be seen in Figure 2, for both the IPM suspensions and total lipid extracts reconstituted into multilamellar liposomes the steady-state anisotropy, 〈*r*〉, of DPH was significantly lower for *ipt1*Δ cells than for *wt* cells, but the values of 〈*r*〉 for IPM and total lipid extracts were similar for each strain.

In conclusion, the results suggest that the lack of M(IP)_2_C in *ipt1*Δ cells and/or the accumulation of MIPC [35,36] may lead to an alteration of the global fluidity of cellular membranes.

### 3.3. Plasma Membrane Fluid Domains Present Similar Properties in *wt* and *ipt1*Δ cells

Electrochromic dyes, such as di-8-ANEPPS, are suitable to report sterol-related properties [50,61]. In fact, the partition of another probe from ANEPPS family, di-4-ANEPPS, for the *l_o_* phase formed in the presence of ergosterol is higher (~2.1×) than for the corresponding *l_d_* phase in the same POPC/ergosterol phase coexistence tie-line. The probe also displays a higher fluorescence quantum yield in the *l_o_* phase [62]. On the other hand, as already mentioned, *t*-PnA partitions preferentially towards gel domains [11,53,54]. Thus, complementary information can be retrieved by labelling the membranes with di-8-ANEPPS [59].

Unlike *t*-PnA, di-8-ANEPPS can be used in fluorescence microscopy studies due to its excitation and emission in the visible range and photostability. The imaging techniques enable to directly conclude from the labelling pattern if the probe is homogeneously distributed in the membrane or if brighter regions can be readily distinguished. We also used FLIM, which allows to selectively analyze the fluorescence lifetime for each region of interest (ROI) of the plasma membrane of living cells. In addition to intact cells, GUV reconstituted from plasma membrane lipid extracts pertaining to either *wt* or *ipt1*Δ cells were also used. These reconstituted systems were double-labelled both with di-8-ANEPPS and with Rhod-DOPE. Rhod-DOPE was only used to label GUVs since when added externally to an aqueous solution it will not incorporate efficiently into the lipid bilayer [49]. Rhod-DOPE, due to the unsaturated acyl chains of DOPE moiety, partitions preferentially towards *l_d_* phase versus both gel [63] and ergosterol-induced *l_o_* [62] domains. In a three-phase coexistence situation, it will be mostly excluded from the gel, but still with some partition into the *l_o_* regions of the membrane [63]. Thus, since di-8-ANEPPS labels preferentially the *l_o_* phase and Rhod-DOPE the *l_d_* phase, their distribution in the membrane allows to evaluate the phase behavior of the GUV prepared from the plasma membrane lipid extracts. Both GUV and living cells were analyzed by confocal microscopy and FLIM (Figure 3).

Confocal images of double-labelled GUVs prepared with plasma membrane lipid extracts from *wt* or *ipt1*Δ cells revealed that the fluorescence intensity of Rhod-DOPE and di-8-ANEPPS was not uniform throughout the GUV membrane. There were regions where the Rhod-DOPE signal was brighter, which indicated the presence of *l_d_* domains. The same was observed for di-8-ANEPPS, which suggested the presence of regions where *l_o_* predominates. Probe distribution throughout the membrane was clearly heterogeneous and despite the existence of regions where either the Rhod-DOPE or the di-8-ANEPPS signal was brighter it was not possible to accurately determine the relative proportion of *l_d_* and *l_o_* since these probes label most of the membrane, though with different intensity. Eventually, the size of the domains may be below the resolution of confocal fluorescence microscopy. Nonetheless, there were regions that were not labelled at all. These dark regions reveal lipid domains from which both probes were excluded. Both these probes are efficiently excluded from gel domains in coexistence with *l*_d_ and *l*_o_ domains but have different distribution between *l*_d_ and *l*_o_ phases [49,64,65]. Therefore, those dark regions that are simultaneously unlabeled by both probes likely correspond to rigid gel domains, in agreement with the results obtained from the fluorescence spectroscopy of *t*-PnA (Figure 1).

FLIM experiments were carried out both on GUV and living yeast cells to study in more detail the biophysical properties of the plasma membrane regions probed by di-8-ANEPPS in *wt* versus *ipt1*Δ cells. In FLIM, the microscopy image generated is based on the fluorescence lifetime rather than fluorescence intensity. Electrochromic dyes such as ANEPPS display a complex relaxation kinetics, which translates into a multiexponential fluorescence decay. It is highly dependent on the density of dipoles and dielectric constant of the membrane interfacial region, polarity properties that can be strongly affected by the presence of sterols [61,66]. In the case of cells, ROI corresponding to the plasma membrane were selected (Figure 4A–D). The parameters describing the fluorescence intensity decays, namely the normalized pre-exponentials and the fluorescence lifetimes of the two exponential components, as well as the amplitude-weighted and intensity-weighted mean fluorescence lifetimes are presented in Table 2.

The lifetimes of the components describing the fluorescence decays obtained in GUV and in living cells are identical (Figure 4E,F) though their amplitudes are not (Figure 4G). Nonetheless, when comparing the two types of cells, *wt* and *ipt1*Δ, the decay parameters obtained were all very similar with no statistical differences found either in the plasma membrane of living cells or in reconstituted GUV, with a percent variation of less than 2%, if at all. These results imply that the properties of the membrane reported by di-8-ANEPPS were not significantly affected by altering the composition of complex sphingolipids.

To further confirm that the properties of ergosterol-enriched domains are not affected by changes in the sphingolipid profile, the ratiometric value *R*_ex_ (420 nm/520 nm), which has a linear relationship with the membrane dipole potential [50], was determined from the excitation spectra of di-8-ANEPPS in the IPM of *S. cerevisiae* (Figure 4H). This parameter was not significantly different between *wt* and *ipt1*Δ cells, corroborating the previous conclusion.

The 〈τ〉 of Rhod-DOPE was also measured in GUVs reconstituted from lipids of *wt* cells yielding a value of 2.36 ± 0.02 ns, whereas in those prepared from lipids of *ipt1*Δ cells a value of 2.45 ± 0.04 ns was obtained. These values lay within those measured in phosphatidylcholine/ergosterol lipid bilayers, where a longer fluorescence lifetime points to a more fluid environment [62]. Thus, the fact that the fluorescence lifetime of Rhod-DOPE in GUVs made from plasma membrane lipids is slightly longer for *ipt1*Δ cells than for *wt* cells, suggests a more fluid membrane in the deletion mutant, in agreement with the results obtained with the probe DPH.

### 3.4. The Lateral Organization of Pma1p But Not Can1p Is Dependent on the Sphingolipid Profile

In order to evaluate the impact of changing sphingolipid composition of *S. cerevisiae* plasma membrane in the organization of MCC and MCP, the time-resolved fluorescence properties of Can1p tagged with GFP (Can1p-GFP) and Pma1p tagged with mRFP (Pma1p-mRFP) were studied by FLIM both in living *wt* and *ipt1*Δ cells. As already mentioned, Can1p is found in MCC domains that co-localize with ergosterol staining by filipin, whereas Pma1p is the hallmark of MCP which seems to be intimately related with sphingolipids, but with no direct evidence for sphingolipid enrichment [3].

The fluorescence decay parameters of the proteins mentioned above were collected for both *wt* and *ipt1*Δ cells (Figure 5 and Appendix A). The decays are bi-exponential as expected when exciting GFP using a wavelength outside the absorption band of the protonated form and detecting emission above 500 nm [67,68,69]. Moreover, the lifetime and amplitude values obtained for each component of the fluorescence decay are in line with those reported in the literature [67,68,69]. The mean fluorescence lifetime obtained for GFP suggests that the protein is close to a medium with a high refractive index [70] as anticipated considering the membrane location of Can1p. For Can1p-GFP no differences were found between *wt* cells and *ipt1*Δ cells. On the contrary, for Pma1p-mRFP significant differences in the fluorescence decay parameters were found between the two strains studied (Appendix A). A statistically significant decrease was also observed in the intensity-weighted mean fluorescence lifetime, from 2.09 ± 0.04 ns in *wt* cells to 1.96 ± 0.07 ns in *ipt1*Δ cells (Figure 5B). This corresponds to, approximately, a 6% variation. This alteration reflects a different environment surrounding the protein in the two strains, which could be related to changes in the membrane refractive index, or to alterations in the pH or local ion concentration in the vicinity of the protein. Considering, e.g., that the refractive index of a lipid membrane varies with the length and packing of the lipid hydrophobic chains [71] and also with polar headgroup chemistry [72], the change obtained in the mean fluorescence lifetime is particularly relevant since it would represent a variation equivalent to ~0.03 in terms of refractive index, or an increase of 20% (*v*/*v*) of glycerol in a water/glycerol mixture for a GFP in solution [70], which, in turn, is also comparable e.g., to a change of about 6 carbons in the length of a fatty acid chain monolayer [71].

The heterogeneity of Can1p and Pma1p distribution in the plasma membrane of *S. cerevisiae* was also assessed (see Section 2.7.2) to further evaluate the possible influence of the complex sphingolipids in the yeast membrane compartments MCC and MCP. The distribution heterogeneity was determined by confocal fluorescence microscopy [15,23], as described in the experimental section. These fusion proteins have been used previously to visualize and detect alterations in both membrane compartments in *S. cerevisiae* [15,23], or similar proteins, e.g., Pma1p-GFP in other fungi such as *Neurospora Crassa* [73] or Pma1p tagged with hemagglutinin in *C. neoformans* [28].

In Figure 6, representative confocal images for the two strains labeled with either Pma1p-mRFP or Can1p-GFP are shown, as well as values obtained for the heterogeneity of their distribution in the plasma membrane for all the replicates. The fact that the heterogeneity values for Can1p and Pma1p distribution in the plasma membrane in *wt* cells are very similar, 0.68 ± 0.10 and 0.66 ± 0.12 respectively, can be due to the reported complementarity of the two membrane compartments. The heterogeneity of Can1p distribution is only slightly—if at all—affected by 3% in *ipt1*Δ cells relatively to *wt* cells (Figure 6A). In contrast, the heterogeneity of Pma1p distribution in the plasma membrane is 15% higher in *ipt1*Δ cells than in *wt* cells (Figure 6B), a prominent change between strains especially when compared to the one found for Can1p. Interestingly, the quantification of fluorescence intensity at the plasma membrane (see Appendix A), reveals that Can1p-GFP has slightly higher average intensity in *ipt1*Δ plasma membrane than in the *wt*, whilst for Pma1p-mRFP, the intensity is significantly weaker in *ipt1*Δ cells. This difference in fluorescence intensity between strains indicates a change in the amount of protein at the plasma membrane, since the fluorescence quantum yield of Pma1p-RFP is only marginally lower in *ipt1*Δ cells, according to the measured amplitude-weighted mean fluorescence lifetimes (Appendix A) and pinpoints a potential defect in Pma1p accumulation at the plasma membrane in *ipt1*Δ cells. The mechanism underlying the lower Pma1p-mRFP levels at the plasma membrane of *ipt1*Δ cells can be related to alterations in the protein traffic or turnover or protein expression levels.

## 4. Discussion

Sphingolipid-enriched gel domains in the plasma membrane of yeast [11] have been proposed to act as diffusion barriers [9] possibly contributing to the stability of the plasma membrane compartments [48] and to the separation of mother and daughter cell components during *S. cerevisiae* cell division that occurs at the budding neck [74]. Another possible function might relate to cell viability in the absence of a cell wall since in the filamentous fungus *N. crassa*, a much tighter packing of sphingolipid-enriched domains occurs in the cell wall-less slime strain when compared to the *wt* strain under similar growth conditions [59] and membrane compartmentalization of lipids and proteins is interconnected to the cell wall and responds to cell wall stress [4]. Although it has been shown that sphingolipids are crucial for the formation and properties of such gel domains [10,11], it is still unclear if these domains contain a major sphingolipid class or if they are composed by a mixture of different sphingolipids. M(IP)_2_C is the most abundant complex sphingolipid in *S. cerevisiae*, particularly at the plasma membrane [35,75], but experimental support allowing the unequivocal identification of the components of gel domains is missing.

Considering the numerous relations between sphingolipids and/or C26 very long chain fatty acids with Pma1p [27,76], it is reasonable to predict that sphingolipid-enriched regions (gel domains) and MCP may co-localize at the yeast plasma membrane, but no direct evidence to confirm this hypothesis is available. Moreover, the impact of the sphingolipid headgroup on Pma1p distribution and microenvironment in the plasma membrane of *S. cerevisiae* has not been elucidated.

Recent observations support the hypothesis that MCC domains are ergosterol-enriched [7]. However, it is still unknown if complex sphingolipids, particularly M(IP)_2_C, play a direct role in the organization of the MCC [3,15].

To address these questions, *wt* cells and *ipt1*Δ cells were compared regarding plasma membrane biophysical properties and distribution and organization of proteins referring to distinct membrane compartments. The deletion of the *IPT1* gene, which leads to the lack of M(IP)_2_C and its replacement by MIPC [35,36], allows to evaluate how changing the polar headgroup of complex sphingolipids influences the different plasma membrane lipid domains, namely, the sphingolipid-enriched gel domains, and also how it impacts on protein distribution and microenvironment. Thus, we aimed at establishing a more direct correlation between sphingolipid-enriched domains and MCP or MCC.

We showed in this work that the formation of gel domains was not hampered by the absence of M(IP)_2_C as the relative abundance of these domains in the plasma membrane was similar for both *S. cerevisiae wt* cells and *ipt1*Δ cells. In fact, in *ipt1*Δ cells, plasma membrane gel domains are even more rigid than those found in *wt* cells, as reported by *t*-PnA time-resolved fluorescence. This can be rationalized considering that M(IP)_2_C is a constituent of gel domains and in *ipt1*Δ cells is replaced by MIPC in those domains. The lack of a second inositol phosphate group makes the polar head of MIPC smaller and less charged than that of M(IP)_2_C. This may allow a tighter packing of these complex sphingolipids due to less electrostatic repulsion and/or steric hindrance at the headgroup level, which may result in a tighter packing and a more rigid environment at the hydrophobic membrane core. A similar proposal has been made to explain the tighter packing of gel domains found in the plasma membrane of *scs7*Δ cells, which lack the 2-OH group in the fatty acyl chain of all sphingolipids, when compared with *wt* cells [11]. On the other hand, the polar headgroups of M(IP)_2_C and MIPC may confer different average transversal positions to the sphingolipid amide group and sphingolipid backbone hydroxyl groups, which leads to distinct H-bonding network at the membrane surface [77] affecting the compactness of gel domains. The biophysical differences reported in this work between *wt* and *ipt1*Δ cells suggest that M(IP)_2_C is an important component of gel domains in *wt* cells. If sphingolipid-enriched domains were mainly composed by MIPC only, and independent of M(IP)_2_C, then in *ipt1*Δ cells, which have a higher concentration of MIPC in the plasma membrane [46,78], we would observe an increase in the abundance of sphingolipid-enriched domains, but no changes in their packing. However, what we found is the opposite, i.e., an increased packing and either no changes or a very small decrease in the abundance of sphingolipid-enriched domains. Thus, in *ipt1*Δ cells the sphingolipid-enriched domains are probably enriched in MIPC, as compared to *wt* cells. This can be due either to a direct effect, i.e., the absence of M(IP)_2_C and its replacement by MIPC or an indirect effect that results from changing the balance of interactions between sphingolipids, including IPC, or between sphingolipids and other membrane lipids.

The loss of the *IPT1* gene leads to an increase in calcium tolerance [36,79], and differential drug resistance/sensitivity phenotypes [80], or an impaired uptake of miconazole [81,82]. Such results indicate that M(IP)_2_C differentially affects membranes permeability or the activity of membrane proteins [80], which suggests that different direct/indirect mechanisms may be at play. It has also been suggested that in *ipt1*Δ *S. cerevisiae* cells, lowered levels of M(IP)_2_C or altered levels of other sphingolipids may regulate the activity of some ABC transporters such as Pdr5p and Yor1p, which have been associated with controlling phospholipid content or distribution at the plasma membrane [83,84]. Thus, alterations of membrane components other than sphingolipids could be responsible, e.g., for the decreased global fluidity of the plasma membrane in *ipt1*Δ cells. Sphingolipid-enriched gel domains have been proposed to serve as diffusion barriers [74], which may account, e.g., for the slow diffusion of plasma membrane proteins in yeast. Thus, the alteration in their rigidity may influence the diffusion properties of several membrane components in the plasma membrane [85], conformational freedom of proteins residing in these domains [86] or the partition of proteins in/out of these domains, namely glycosylphosphatidylinositol-anchored proteins [11].

While changing the sphingolipid headgroup has a measurable impact on the compactness of gel domains, it has no obvious effect on the lipid-dependent dipolar properties of sterol-enriched domains. With di-8-ANEPPS, our results for both fluorescence lifetime and membrane dipole potential measurements, were nearly identical for *wt* and *ipt1*Δ cells, regardless of the model system studied—GUVs, living cells or IPM. This suggests that those biophysical properties, which are strongly dependent on sterol and sterol-lipid interactions [51,61], are less dependent on the major sphingolipid headgroup. In *erg6*Δ cells, which have an altered sterol profile, but no noticeable changes in sphingolipid profile when compared to *wt* cells [87,88], the rigidity of the gel domains is not changed [11], but the fluorescence lifetime of a sterol sensitive dye is significantly different from *wt* cells [51]. This behavior is opposite to that found for *ipt1*Δ cells where the rigidity of the gel domains is increased but the fluorescence properties of di-8-ANEPPS remain unchanged when compared to *wt* cells. Hence, sphingolipid-enriched and ergosterol-enriched regions seem to be to a certain extent independent in the plasma membrane of yeast. Those results thus support the recent observation that sphingolipids and ergosterol do not predominate in the same membrane leaflet [89]. Nonetheless, the occurrence of sphingolipid-enriched domains in the outer leaflet does not rule out the presence of ergosterol molecules in the inner leaflet of MCP.

Regarding the protein markers of membrane compartments, it was found that Pma1p plasma membrane distribution heterogeneity and fluorescence lifetime are changed in *ipt1*Δ cells when compared to *wt* cells, whereas the same parameters remain unaltered for Can1p. Since Pma1p- and Can1p-specific membrane compartments have been described as spatially separated [1,3], and considering that changes were observed in the biophysical properties of sphingolipid-enriched domains but not in ergosterol-enriched domains, the results obtained in this work for Pma1p and Can1p support an independent compartmentalization of sphingolipid-enriched and sterol-enriched regions, and that these regions could be lipid domain counterparts of the MCP and the MCC, respectively. Currently, it is not certain which lipids compose either MCP or MCC apart from the lower ergosterol content of the former and the higher ergosterol content of the latter [15,23,86]. However, by using a lipidomic approach with styrene-maleic-acid polymers (SMAPLs) to determine which lipids surround Can1p and Pma1p, and other MCC or MCP specific proteins (periprotein lipidome), Klooster et al. [86] demonstrated that in fact the sphingolipid/ergosterol ratio is much higher in the MCP than in the MCC, while they do not differ much in periprotein phospholipid composition. The distribution pattern of Pma1p as isolated *foci* in the plasma membrane has been associated to the honeycomb-like hexagonal lattices observed on the plasma membrane of *S. cerevisiae* [1]. Moreover, it has been suggested that the lipids contained in these structures should be very compact and display a gel arrangement [1]. The results here presented, and considering the body of knowledge on the relation between sphingolipids and sterols in MCP organization, e.g., [1,26,27,28,29,30,31,32,33,86], are consistent with the presence of gel-forming sphingolipids in the composition of the MCP, providing the most simple explanation for the concomitant changes here observed in gel domains packing and Pma1p-mRFP, but not Can1p-GFP, distribution and fluorescence lifetime. The different fluorescence lifetime of Pma1p-mRFP in *wt* vs. *ipt1*∆, cells reflects a different environment for the protein. This could be due to a different refractive index in the vicinity of the protein, which could be attributed to its relocation into different domains or changes in the thickness and packing of those domains, or it could also be due to misfunction of Pma1p leading to alterations of pH and ion concentration in the vicinity of the protein. This is a subject that certainly deserves further investigation. Our data strongly suggest that the MCC does not contain gel domains, since an alteration of their compactness did not affect the properties measured for Can1p. However, it should be noted that Can1p MCC partitioning is less affected than other MCC transporters by sphingolipid stress [3].

The labelling of GUV reconstituted from plasma membrane lipid extracts of *wt* cells or *ipt1*Δ cells with both di-8-ANEPPS and Rhod-DOPE revealed a complex phase behavior where three distinct types of domains can be distinguished and related to a three-phase coexistence (*l*_d_/*l*_o_/gel). DPH steady-state fluorescence anisotropy was lower in the IPM of *ipt1*Δ cells, pointing to a lower global order of the membrane, which seems to contradict the higher order of the gel domains. Therefore, it seems that the remainder of the membrane is more fluid. In particular, there could be a small increase in the relative abundance and/or fluidity of *l_d_*-like domains. Upon *IPT1* deletion no alterations in *l*_o_-like domains were apparent, since fluorescence decays of di-8-ANEPPS yielded similar components lifetimes and amplitudes in *ipt1*Δ and *wt* cells, and the gel domains became more rigid (and maybe slightly less abundant). Phytoceramide, the backbone of complex sphingolipids in yeast, can recruit up to 3 glycerophospholipid molecules per molecule of phytoceramide to form rigid gel phases [48]. It is possible that each M(IP)_2_C molecule can recruit a higher number of glycerophospholipid molecules, not only due to its double negative charge that could interact with the positive choline moiety of phosphatidylcholine, but especially because of the large size of its polar headgroup. MIPC, which is less charged and contains a smaller headgroup compared to M(IP)_2_C, would recruit a lower number of glycerophospholipid molecules than M(IP)_2_C. Thus, in *ipt1*Δ cells, where M(IP)_2_C is missing and MIPC accumulates instead, gel domains would be more rigid due to the lower content in glycerophospholipids. On another hand, these low melting temperature lipids would become more available for the formation of disordered domains, contributing to the higher global fluidity of the membrane in *ipt1*Δ cells. Since gel domains are more rigid in *ipt1*Δ cells, this would suggest that liquid domains are more disordered in this strain. It was previously confirmed with DPPC and POPC membranes with and without cholesterol, labelled with Rhod-DOPE at different probe/lipid ratios [62,90], that the higher the 〈τ〉 values for Rhod-DOPE the more disordered are the membrane regions into which the probe is incorporating. Therefore, the fact that 〈τ〉 of Rhod-DOPE in GUVs reconstituted from lipids of *wt* cells is lower than those prepared from lipids of *ipt1*Δ cells supports the hypothesis that liquid regions in the plasma membrane of *ipt1*Δ cells are more disordered when compared to *wt* cells. Additionally, considering the recent results on the Pma1p periprotein lipidome [86], where this region of the membrane is sphingolipid-enriched, but there seems to be some ergosterol present, one might hypothesize that in *ipt1*Δ cells the sphingolipid-enriched domains /Pma1p periprotein lipidome become even more depleted in ergosterol, which could also lead to an increased rigidity of the domains, because ergosterol may actually fluidify these domains [11]. However, it is unlikely that the changes are due to ergosterol, because there were no changes in the fluorescence parameters of di-8-ANEPPS in *ipt1*Δ cell membrane and, in the *erg6*Δ cells strain lacking ergosterol mentioned above, the rigidity of the gel domains was not different from the *wt*.

This work points to a role of the polar headgroup of complex sphingolipids in the organization of Pma1p in the plasma membrane of *S. cerevisiae*. Also, in *C. neoformans* the complex sphingolipid headgroup has been related with Pma1p oligomerization and transport [28]. By regulating the rigidity/order of gel domains, sphingolipid polar headgroups may thus contribute to control Pma1p stability, distribution, and oligomerization in the plasma membrane. However, a specific interaction of M(IP)_2_C, having an extra inositol phosphate group as compared to MIPC, with the protein cannot be ruled out. If the lipid annulus surrounding the protein is different, e.g., in terms of number of lipid rings, the fraction of annular or non-annular complex sphingolipid may change, resulting in a different overall packing within the sphingolipid-enriched domains.

The present work shows how the headgroup structure of complex sphingolipids can be an important determinant in the organization of plasma membrane lipids and proteins in *S. cerevisiae*. Recently, we showed that the polyene antifungal nystatin forms active pores in membranes lacking any sterol, but containing gel domains [12], challenging the classical model which describes the molecular mode of action of nystatin through an ergosterol-dependent mechanism. Moreover, the action of other antifungal drugs has been shown to involve an interplay with sphingolipids that is conditioned by their headgroup structure. *S. cerevisiae* cells defective in M(IP)_2_C are resistant to zymocin [37]. Moreover, zymocin requires an active Pma1p proton pump to exert its toxicity. These observations can be related to our results supporting a close relation between MCP and sphingolipid-enriched domains. The sphingolipid headgroup also influences the antifungal activity of the lipopeptide syringomycin E [38]. Fungi lacking M(IP)_2_C, and accumulating MIPC, become resistant to syringomycin E. Syringomycin E acts on the membrane forming channels causing K^+^ efflux and Ca^2+^ influx. The complete absence of M(IP)_2_C hampers syringomycin E toxicity. Since in *ipt1*Δ cells, gel domains become more tightly packed, this might hinder the insertion of syringomycin E acyl chain into the membrane. An independent study supporting the importance of sphingolipid-dependent biophysical properties shows that *scs7*Δ cells, where sphingolipid-enriched domains are also more rigid than in *wt* cells [11], are also more resistant to the action of syringomycin E. Since the lack of M(IP)_2_C leads to a tightening of gel domains, as well as to an altered distribution of Pma1p, it is possible that membrane proteins, alongside lipid composition, are also important for the susceptibility to syringomycin E. This is further supported by the observation that Pma1p proper functioning is crucial for the virulent activity of pathogenic fungi [28,91,92]. The apparent requirement of intact MCP/M(IP)_2_C domains for the action of syringomycin E is supported by the fact that cumulative conditions of MIPC accumulation and absence of M(IP)_2_C lead to resistance; yet higher levels of MIPC, when M(IP)_2_C is still produced, do not change the sensitivity phenotype [93]. A better understanding of sphingolipids role in yeast plasma membrane organization, particularly, in domain formation, may thus be an important contribute to clarify antifungal molecular targets and mode of action.

Gournas and collaborators showed that sphingolipids are required for the slower diffusion of Can1p in the MCC [8] and, presented different hypothesis for the still unknown mechanism [3]. Our results seem to rule out the hypothesis that M(IP)_2_C is directly required for the conformation-dependent anchoring of the protein transporters in the MCC, since its absence does not alter any measured property of Can1p. Also, in this work we observed diverse membrane reorganizations upon *IPT1* deletion, including changes in global plasma membrane fluidity, and yet no significant alterations were detected for Can1p, which seems to discard the hypothesis that the partitioning of Can1p into the MCC could be indirectly affected by a general alteration of plasma membrane organization. Our results did not confirm neither reject the possibility that sphingolipids depletion could lead to a concomitant decrease of ergosterol levels in eisosomes, since ergosterol levels are similar in both *ipt1*Δ cells and *wt* cells and we show that some of the sterol-dependent properties are not significantly changed in the plasma membrane of *ipt1*Δ cells when compared to *wt* cells. However, another scenario is still possible, where sphingolipid-enriched domains act as diffusional barriers and are neighboring the MCC, thus preventing movement of Can1p. The change in sphingolipid levels may induce an alteration of the relative arrangement of yeast compartments, indirectly affecting those that are not sphingolipid-enriched.

## 5. Conclusions

There are still many unaddressed questions regarding lipid–protein interactions in the plasma membrane of fungi. Disclosing the mechanism of membrane compartments formation, their lipid composition and interrelations will hopefully lead to significant advances in membrane biology, but their investigation is also driven by the importance of some of these compartments in the pathogenicity of clinically relevant fungal species.

The identification of gel domains under physiological conditions was an important finding but the specific molecular constitution of those domains is unknown. Four different fluorescent probes, *t*-PnA, DPH, di-8-ANEPPS, and Rhod-DOPE, were used to label yeast plasma membrane, in living cells and IPM and in reconstituted systems, namely GUV made from plasma membrane lipid extracts. The set of probes employed yields complementary information that enables a comprehensive picture of the biophysical changes taking place at the plasma membrane upon the loss of M(IP)_2_C synthesis ability. With this work, it became clear that M(IP)_2_C is important for the regulation of gel domains in the plasma membrane of yeast, since in cells unable to synthesize this complex sphingolipid these domains are more rigid. In the future, co-localization studies with Pma1p-mRFP, and Can1p-GFP-containing *wt* cells [23], *ipt1*Δ cells and other strains with mutations in sphingolipid (and sterol) metabolism will provide invaluable insights to further understand the intricate relation between membrane lipid composition, biophysical properties, and membrane compartments organization and function in yeast.

Our results also help to clarify the interrelation between sphingolipids, M(IP)_2_C in this case, and Pma1p or Can1p. Diverse results in the literature point to a dependence of Pma1p proper sorting with intact sphingolipid synthesis. Here, we demonstrate a direct correlation between sphingolipid polar headgroup and protein distribution. Pma1p distribution is more heterogeneous in *ipt1*Δ cells, whereas Can1p distribution, if affected at all, becomes slightly more homogeneous. This suggests that M(IP)_2_C is directly involved in the organization of MCP, but possibly not of MCC. The concomitant alteration in the rigidity of gel domains and Pma1p distribution supports a close communication between gel domains and MCP. On the other hand, MCC and ergosterol-enriched domains did not undergo any noticeable changes upon a major alteration of complex sphingolipid profile, which corroborates that MCC and ergosterol co-localize, further supporting that M(IP)_2_C is not an important constituent of ergosterol-enriched domains.

## Figures and Tables

**Figure 1 biomolecules-10-00871-f001:**
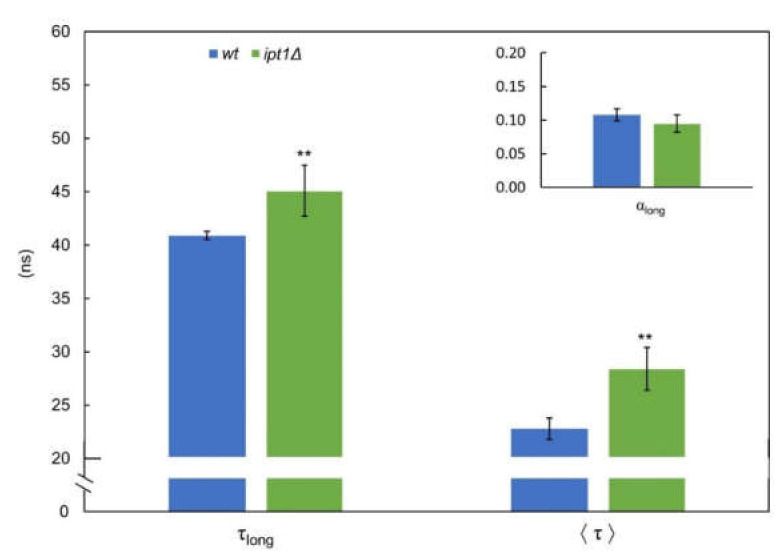
The gel lipid domains of the plasma membrane of intact *S. cerevisiae* cells are more compact in *ipt1*Δ than in *wt* cells. The fluorescence intensity decay of the probe *t*-PnA in *S. cerevisiae* cellular suspensions was obtained at 24 °C, as described in the “experimental procedures”. The long-component lifetime (τ_long_) and normalized amplitude (α_long_; see inset) and the mean fluorescence lifetime (〈τ〉) are shown. The values are the mean ± S.D. of at least four biological replicates. ** *p* ≤ 0.01.

**Figure 2 biomolecules-10-00871-f002:**
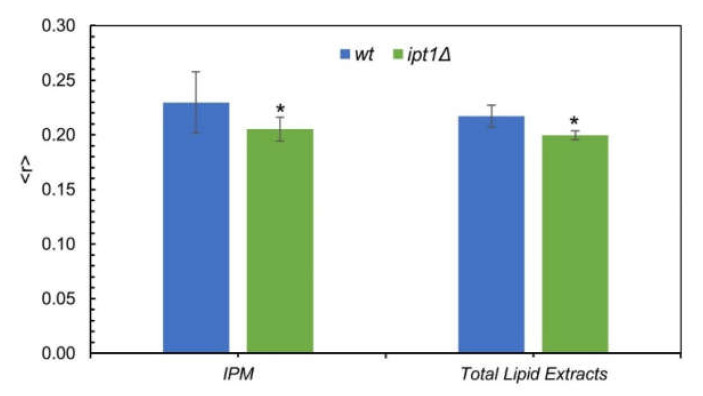
The global order of the plasma membrane of *S. cerevisiae* is higher in *wt* than in *ipt1*Δ cells. The diphenylhexatriene (DPH) steady-state fluorescence anisotropy at 24 °C was obtained as described under “experimental procedures” for *wt* and *ipt1*Δ cells isolated plasma membrane (IPM) and multilamellar vesicles (MLVs) made from total lipid extracts obtained from cells in mid-exponential phase. The values are the mean ± S.D. of at least four biological replicates. * *p* ≤ 0.05.

**Figure 3 biomolecules-10-00871-f003:**
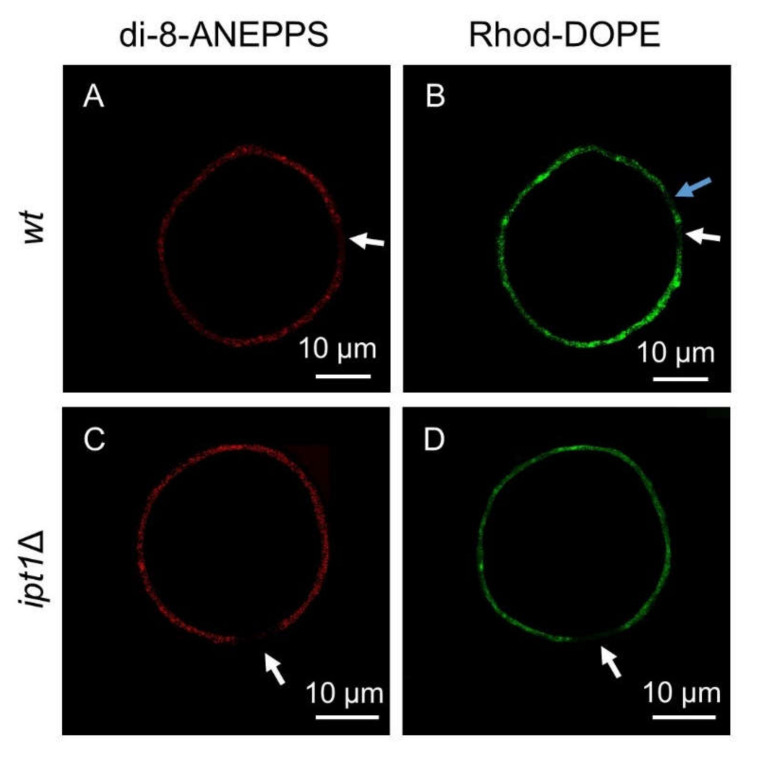
*S. cerevisiae* plasma membrane lipids have complex phase behavior. Representative confocal microscopy images of giant unilamellar vesicles (GUVs) reconstituted from plasma membrane lipid extracts obtained from *wt* cells (**A**,**B**) or *ipt1*Δ cells (**C**,**D**) double-labelled with both di-8-ANEPPS and Rhod-DOPE. Images from di-8-ANEPPS channel (**A**,**C**) and Rhod-DOPE channel (**B**,**D**) are shown. White arrows indicate regions that were poorly labelled by both probes. The blue arrow points to a region that was poorly labelled by Rhod-DOPE but not at all by di-8-ANEPPS.

**Figure 4 biomolecules-10-00871-f004:**
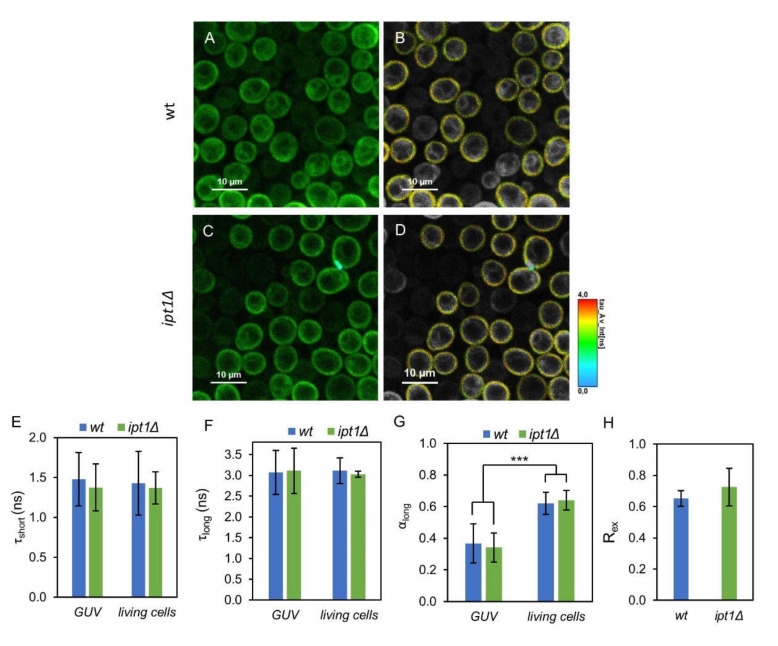
Di-8-ANEPPS parameters are identical for *wt* and *ipt1*Δ cells in the different systems studied. Representative fluorescence lifetime imaging microscopy (FLIM) images of living *S. cerevisiae wt* (**A**,**B**) and *ipt1*Δ cells (**C**,**D**) labelled with di-8-ANEPPS. Panels (**A**,**C**) show representative intensity images, whereas in panels (**B**,**D**) the region-of-interest (ROI) selecting the plasma membrane is represented with the lifetime color scale obtained by FLIM. Lifetime components in GUVs obtained from plasma membrane lipids of *wt* and *ipt1*Δ cells and in living cells, (**E**) lifetime of the short component, (**F**) lifetime of the long component and (**G**) amplitude of the long component. The amplitude of the short component is 1-α_long_. The values are the mean ± S.D. of at least three biological replicates with a total of at least 50 GUVs and 400 cells analyzed per strain. ***, *p* ≤ 0.001. In panel (**H**) R_ex_ (420 nm/520 nm), which is linearly dependent on the membrane dipole potential, is shown for IPM suspensions labeled with di-8-ANEPPS (*n* ≥ 3).

**Figure 5 biomolecules-10-00871-f005:**
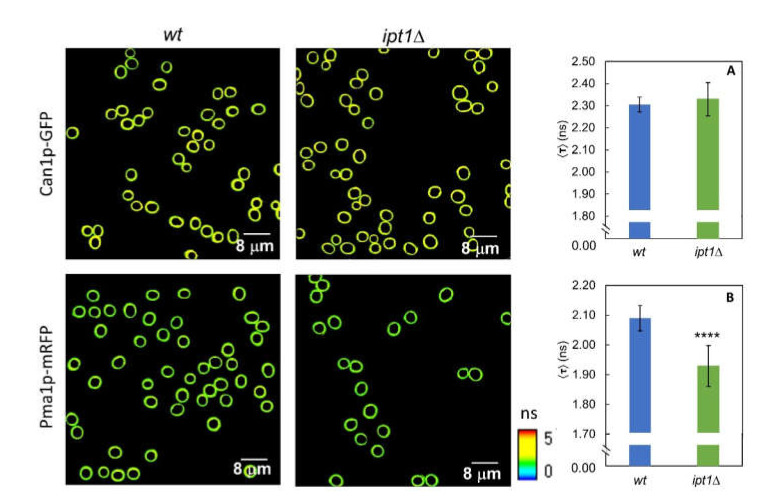
The lack in M(IP)_2_C leads to a change in the microenvironment surrounding Pma1p but not Can1p. Representative FLIM images of ROI comprising *S. cerevisiae wt* cells and *ipt1*Δ cells plasma membrane. Intensity-weighted mean fluorescence lifetime for *S. cerevisiae* cells. (**A**) *wt*-Can1p-GFP and *ipt1*Δ-Can1p-GFP cells and (**B**) *wt*-Pma1p-mRFP and *ipt1*Δ Pma1p-mRFP cells. The values are the mean ± S.D. of three independent biological replicates with a total of ca. 400 cells analyzed for each strain. **** *p* ≤ 0.0001.

**Figure 6 biomolecules-10-00871-f006:**
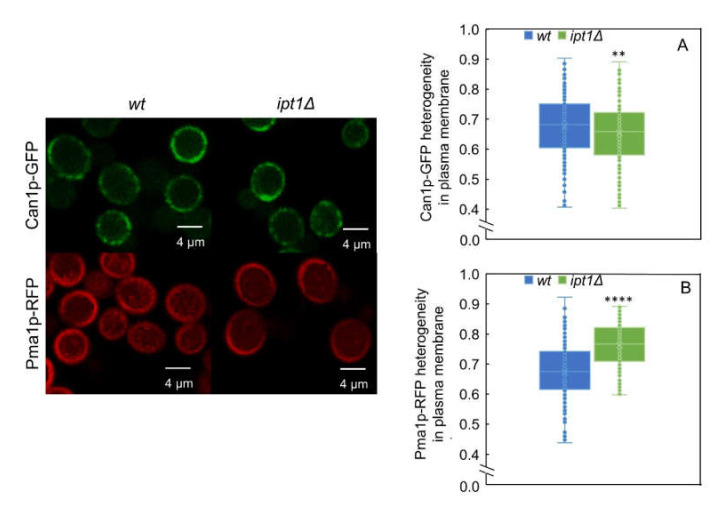
MCP distribution in the plasma membrane is different in *wt* cells and *ipt1*Δ cells. The heterogeneity of Can1p-GFP and Pma1p-mRFP distribution along the plasma membrane was determined for *wt* cells (**A**) and *ipt1*Δ cells (**B**) as described in the experimental procedures. The values are the median ± S.D. of at least four biological replicates with a total of at least 190 cells analyzed per replicate. **, *p* ≤ 0.01; ****, *p* ≤ 0.0001. On the left, representative confocal fluorescence images of living *S. cerevisiae* cells with or without deletion of *IPT1* and expressing either Can1p-GFP or Pma1p-mRFP, as indicated, are shown.

**Table 1 biomolecules-10-00871-t001:** *Saccharomyces cerevisiae* strains used in this work, genotype description, and growth media.

Strain	Genotype	Source	Growth Media
wild-type (*wt*; BY4741)	*MATa; his3* *Δ* *1; leu2* *Δ* *0; met15* *Δ* *0; ura3* *Δ* *0*	EUROSCARF (Frankfurt, Germany)	Synthetic Complete medium (SC)
*ipt1*Δ	BY4741; *YMR272c::kanMX4*	EUROSCARF (Frankfurt, Germany)	SC
*wt*-Can1p-GFP	BY4741; *Ylp211CAN::GFP*	This study	SC *ura^-^*
*wt*-Pma1p-mRFP	BY4741; *Ylp128PMA1:mRFP*	This study	SC *leu^-^*
*ipt1*Δ-Can1p-GFP	BY4741; *YMR272c::kanMX4; Ylp211CAN::GFP*	This study	SC *ura^-^*
*ipt1*Δ-Pma1p-mRFP	BY4741; *YMR272c::kanMX4; Ylp128PMA1::mRFP*	This study	SC *leu^-^*

**Table 2 biomolecules-10-00871-t002:** Fluorescence intensity decay parameters of di-8-ANEPPS incorporated in *S. cerevisiae wt* and *ipt1*Δ cells at 24 °C. The fluorescence intensity decays measured in cells were described by the sum of two exponentials, with amplitudes α_1_ and α_2_ and lifetimes τ**_1_** and τ**_2_**. $\tau_{av}$ is the amplitude-weighted mean fluorescence lifetime and 〈τ〉 is the intensity-weighted mean fluorescence lifetime (Equations (3) and (4)). The values are the mean ± S.D. of more than 50 GUVs from three biological replicates and ca. 400 cells from four biological replicates.

Membrane System	Strain	τ_1_ (ns)	α_1_	τ_2_ (ns)	α_2_	$\bar{\tau }$ (ns)	〈τ〉 (ns)
GUVs	*wt*	1.48 ± 0.33	0.63 ± 0.12	3.07 ± 0.53	0.37 ± 0.12	2.05 ± 0.32	2.35 ± 0.35
	*ipt1*Δ	1.37 ± 0.29	0.66 ± 0.09	3.11 ± 0.54	0.34 ± 0.09	1.95 ± 0.33	2.30 ± 0.37
Living cells	*wt*	1.43 ± 0.40	0.38 ± 0.07	3.00 ± 0.31	0.62 ± 0.07	2.43 ± 0.12	2.70 ± 0.06
	*ipt1*Δ	1.37 ± 0.20	0.36 ± 0.06	3.03 ± 0.07	0.64 ± 0.06	2.43 ± 0.15	2.69 ± 0.07

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
