# Peer review of "Yeast Sphingolipid-Enriched Domains and Membrane Compartments in the Absence of Mannosyldiinositolphosphorylceramide"

_biomolecules, 2020, doi:10.3390/biom10060871_

Round 1

Reviewer 1 Report

Review of “Sphingolipids-enriched domains and membrane compartments in yeast: role of mannosyldiinositolphosphoryceramide”

In the manuscript entitled “Sphingolipids-enriched domains and membrane compartments in yeast: role of mannosyldiinositolphosphoryceramide” Bento-Oliveira and co-authors investigate the effect of the loss of IPT gene, which is required for the synthesis of mannosyl-diinositol phosphoryceramide (M(IP)2C), on plasma membrane biophysical properties and organization in yeast. Using spectroscopy, fluorescence lifetime imaging and confocal microscopy to observe fluorescent membrane probes as well as translational fusion of two plasma membrane-localized proteins the authors describe notably that loss of IPT increases the rigidity of gel-like membrane domains and increase heterogeneity of Pma1p. Results presented by the authors document the implication of sphingolipid polar heads in regulating biophysical properties and organization of the plasma membrane and are of interest. I am however concerned by the interpretation and conclusion drawn by the authors. The manuscript requires important revision.

Here are specific comments:

-The authors affirm that observed phenotype in ipt is due the absence of M(IP)2C. For example, they conclude that increase in rigidity of gel-like domains and increase in heterogeneity of Pma1p is a consequence of the absence of M(IP)2C and that M(IP)2C is a constituent of Pma1p membrane domains. Because iptaccumulates MIPC wouldn’t it be possible that accumulation of MIPC increases the rigidity of gel-like domains, increases the heterogeneity of Pma1p and that MIPC is a constituent of Pma1p membrane domains? Following this possibility M(IP)2C would be absent from Pma1p membrane domains and would not play any role in regulating them.  This would be in contradiction to the main conclusion drawn by the authors. To reflect this possibility, the manuscript should be amended, notably the title and parts dealing with the interpretation of the results, the discussion and the conclusion.

-L347-352. Nothing indicate that dark regions are the same for both probes. To make such statement one would need to observe both probe at the same time, please rephrase. In addition, to state that these dark regions correspond to gel-like domain one would need to observe at the same time a marker specific for gel-like domains. Please rephrase.

-L392-393: Interpretation of the data collected for Rhod-DOPE in GUVs is missing.

-Figure 5: pH and ion concentration modulate fluorescence protein lifetime. I suspect that differences in mRFP lifetime observe for Pma1p comparing WT and iptare due to misfunction of Pma1p in ipt, and therefore alteration of pH and ions concentration in the vicinity of Pma1p, rather than changes in the reflective index as propose by the authors. In any cases, alteration of the membrane thickness by few angstroms does not impact fluorescence lifetime as measured by time-correlated single-photon counting device.

-Figure 6: To pinpoint any potential defect in Can1p-GFP and Pma1p-mRFP accumulation at the plasma membrane the authors should provide quantification of fluorescence intensity at the plasma membrane for both proteins comparing WT and ipt. Fluorescence of Pma1p seems lower in iptthan in WT, does it affect measurement of Pma1p heterogeneity?

-Overall writing of the manuscript could be improved.  For example, lines 100-102 could be concatenated and rephrased as follow “The formation of the terminal complex sphingolipid M(IP)2C from MIPC is catalysed by an inositol phosphotransferase encoded by the IPT1 gene”. Also, clear descriptions of the parameters measured and their meaning in the context of these experiments would make the manuscript understandable by a broader audience.

Reviewer 3 Report

please see the attached file (comments-suggestions)

Reviewer 4 Report

All experiments in the manuscript were carried out using only wt and ipt1KO stains to compare effects of loss of ipt1 on membrane properties. This study would be benefited with an inclusion of relevant positive control strains that may display expected positive results, which helps the authors conclude the role of IPT1 gene on membrane properties and gain more confidence with their conclusion. Furthermore, the research presented here lacks mechanistic insights on how loss of IPT1 gene affect other constituents at the membrane or its vicinity to induce the observed phenotypes or how normal Ipt1 function with these components for membrane homeostasis. Taken together, the reviewer does not think this manuscript is of sufficient quality, novelty, or importance to warrant publication in this journal.

Introduction

Line 53: references (9,10) need to move to the end of the sentence.

Lines67-69: The deletion of 68 either eisosomal or MCC constituent proteins compromises the formation of intact MCC/eisosome 69 domains, and influences the distribution of ergosterol in the plasma membrane [2]. è double check the logic and grammar of the sentence.

Line 89: need a space between time and the reference.

Line 102: “mutation”  è consider to switch it into deletion?

Line 111: we have, carried =è remove comma.

Results

Lines 295-297: it would have been better if this sentence included the conclusion of this set of experiment as shown in the subtitle.

Lines 313-314: based on the literature, ipt1 KO cells lack M (IP)2C. However, the authors for the present work must check the level of this lipid for their KO cells as a control experiment. Or any alternative evidence that these KO cells have abnormally high level of MIPC needs to be provided.

Line 320: on the other hand?

Line 322: a space between they reporter and the reference.

Line 328: after (ROI)è need a preposition.

Line 339-343 è cite the figure 3.

Line 382 è some information within the parenthesis would need to be removed.

Line 384: percentual variation è percent variation?

Line 451: if it all è if at all.

Discussion

Lines 489-490: one can not conclude that the observed phenotype (more rigid) are purely due to the loss of IPT1 since the loss could lead to indirect or side effects. Therefore, the authors need a complementation assay in which an expression plasmid that express functional IPT1 in the background of ipt1KO in order to see if this expression suppress all the negative phenotypes. It would be also helpful to overexpress IPT1 in a wt background to see the effect on membrane properties.

Lines 490-494: it may be that the resulting rigidity is attributed to an elevated ergosterol levels in the mutant strain. Recommend to test this possibility.

Lines 500-501: it is not that the reviewer disagrees to this idea, but the reader would be attracted to know mechanistic insights on how loss of IPT1 gene affect other constituents at the membrane or its vicinity to induce the observed phenotypes or how normal Ipt1 function with these components for membrane homeostasis.

Round 2

Reviewer 1 Report

I Thank the authors for their thorough point-by-point reply. The authors significantly improved the quality of their manuscript by notably amending the interpretation of the results, the discussion and by providing precision on the methodology.

The authors answer all my previous comments and concern. I do not have any additional comments to further improve the manuscript and congratulate the authors for their efforts and overall work presented in the manuscript.

Best wishes

Reviewer 2 Report

The authors satisfactory addressed all my critical comments.

Reviewer 4 Report

All suggested changes by the reviewer were adopted and appropriately revised by the authors, leading to the current manuscript that is acceptable for publication in the journal.